# Custom-Made Direct Metal Laser Sintering Titanium Subperiosteal Implants in Oral and Maxillofacial Surgery for Severe Bone-Deficient Patients—A Pilot Study

**DOI:** 10.3390/diagnostics12102531

**Published:** 2022-10-19

**Authors:** Alexandru Nemtoi, Vlad Covrig, Ana Nemtoi, George Stoica, Ruxandra Vatavu, Danisia Haba, Irina Zetu

**Affiliations:** 1Department of Biomedical Sciences, Faculty of Medicine and Biological Sciences, “Stefan cel Mare” University, 13 Universitatii Str., 720229 Suceava, Romania; 2Department of Oral and Maxillofacial Surgery, Faculty of Dental Medicine, “Gr. T. Popa” University of Medicine and Pharmacy, 16 Universitatii Str., 700115 Iasi, Romania; 3Department of Dentistry, Faculty of Medicine and Pharmacy, Dunarea de Jos University, 47 Domneasca Str., 800008 Galati, Romania; 4Department of Morpho-Functional Sciences I, Faculty of Medicine, “Gr. T. Popa” University of Medicine and Pharmacy, 16 Universitatii Str., 700115 Iasi, Romania

**Keywords:** subperiosteal implant, CBCT, direct metal laser sintering, titanium implants

## Abstract

Background: Nowadays, a combination of classical subperiosteal implant designs with 3D imaging and printing allows one to reduce treatment time and provides support for fixed prostheses in cases where other techniques do not provide satisfactory results. This study aims to present a digital technique for the manufacturing of custom-made subperiosteal implants and what complications might appear after this type of surgery. Methods: Sixteen patients treated with a custom-made DMLS titanium subperiosteal implant during the period between October 2021 and February 2022 were enrolled in the study. Orthopantomography (OPT) and cone-beam computer tomography (CBCT) were recorded for all patients. The measurements taken into account in this study were the fit and stability of implants, duration of surgery, implant survival, and early and late complications. Results: The fit of the implants was extremely satisfactory, with a mean rating of 4 out of 5. The mean duration of the intervention was 86.18 min. At the end of the study, one implant was lost due to insufficient fit and recurrent, untreatable infections. Eleven implants (69%) were placed on the maxillary and five (31%) implants were placed on the mandible. Conclusions: Taking this into consideration, custom-made DMLS titanium subperiosteal implants could present satisfactory implant survival and low complication rates.

## 1. Introduction

Dental implants represent the most used type of surgical treatment in treating partially and edentulous patients, with a high percentage of survival and success in the medium and long terms [1,2,3,4]. 

In this situation, the patients need sufficient quantity (height and width) and quality (density) of bone to be able to insert endosseous dental implants. In many situations, the patients do not have enough bone, and the surgeon needs to perform some surgical techniques for bone augmentation before or at the same time as the endosseous dental implant surgery [5]. 

Different surgical protocols have been proposed for bone augmentation to allow for the insertion of endosseous implants. These protocols include onlay/inlay bone grafting [6,7], guided bone regeneration (GBR) with resorbable or non-resorbable membranes [8], ridge split [9], distraction osteogenesis [10], and maxillary sinus augmentation [11,12]. All of these bone surgical procedures use different materials, such as autogenous bone harvested from intraoral/extraoral sites, and homologous, heterologous, or synthetic bone grafts, with the role of restoring bone volume.

However, these surgical procedures are complex, with a high rate of complications which lengthen the time of treatment. Furthermore, these techniques present additional costs to patients. There are some situations with severe atrophy of the jaws, in which bone augmentation surgery is not recommended because of the poor quality of the basal bone.

In the past, there was a possibility of placing a subperiosteal implant in cases of severe bone resorption, but the manufacturing technique was rudimentary for this type of implant and was abandoned in favor of endosseous implants. A subperiosteal implant is a type of implant that is placed under the periosteum, directly on the maxillary or mandibular bone [13,14]. It usually has four transmucosal parts which pass through the mucosa and into the oral cavity, connecting the implant to the fixed prosthesis [14,15].

The first time this type of implant was used was in the early 1940s in Sweden [13,16]. The subperiosteal implants were made from different alloys such as chrome–cobalt or titanium [15,16,17]. In cases of severe bone resorption of alveolar ridges, these implants were usually used by placing them above the bone and were frequently immediately loaded with a fixed prosthesis [13,15,16,17,18,19].

This type of implant received certain popularity for a period of about twenty years [17,18,19,20,21] until they were replaced by endosseous dental implants, as proposed by Branemark [4,22]. The fabrication techniques were very complex for a subperiosteal implant, with considerable discomfort for the patient during preparation. This was usually caused by nonideal adaptation to the region of interest of the implant, higher risk of postoperative infections, and different complications. All of these factors turned attention to the development of different surgical techniques with different types of endosseous implants such as narrow, short, and tilted implants.

At present, oral implantology has evolved tremendously due to digital technology that has improved surgical protocols, especially in the planning stage [23]. The introduction of different techniques in medical practice such as cone-beam computed tomography—CBCT [24,25,26,27]; intraoral scanners [28], along with computer-assisted design/computer-assisted manufacturing (CAD/CAM) software [29], modern materials, and manufacturing technologies, has dramatically changed the field of oral implantology, opening up new directions [30].

This digital period offers new perspectives, such as 3D visualization and printing, and, in particular, direct metal laser sintering (DMLS) [31], which allows the manufacturing of custom-made maxillofacial prostheses [32,33] and even implants [34,35] perfectly adapted to the patient’s specific anatomy.

Nowadays, the possibility of using such modern manufacturing techniques allows us to review some of the past techniques, such as periosteal implants, and gives us the possibility to work with them in a modern and digital way.

This surgical technique offers new solutions for patients with severe bone atrophy or poor quality of bone, such as in the case of certain systemic diseases such as osteoporosis, diabetes mellitus, rheumatoid arthritis, or a local, oral pathology, such as severe periodontitis, and peri-implantitis. With conventional techniques, peri-implantitis does not allow for the insertion of endosseous dental implants to place a custom-made structure to sustain fixed dental restorations [36,37,38].

The purpose of this study is to present a digital technique for the fabrication of custom-made DMLS subperiosteal implants and to present the early and tardive complications which appeared when using these types of implants in patients with severe bone resorption and poor bone quality.

## 2. Materials and Methods

### 2.1. Patient Selection

The information for this study was taken from two private Oral and Maxillofacial Surgery Centers, located in northeastern Romania, for 12 months. This clinical information contained surgical documentation of cases.

In total, 19 patients who had been treated with a custom-made DMLS titanium subperiosteal implant during the period between October 2021 and February 2022 were considered for enrolment in this study. Of these 19 patients, 3 of them did not have complete documentation and were excluded from the study. The final group of patients included 16 patients (9 males and 7 females, aged between 55 and 69 years, mean age of 61.5 years). 

The inclusion criteria for the patients to be taken in the study were patients over the age of 55, treated with custom-made DMLS titanium subperiosteal implants, equilibrate general and oral health, improved oral hygiene, totally and extended partially edentulous with marked bone resorption which does not allow insertion of standard size implants (length ≥ 10 mm), the desire not to benefit from bone regeneration surgery, and availability to participate in postoperative consultations.

The exclusion criteria for the patients to be taken in the study were patients under 55 years, general pathologies, or pharmacological therapies that could contraindicate the intervention (such as immunocompromised patients, uncontrolled diabetes, tumors of the head and neck, or treatment with bisphosphonates), inadequate oral hygiene, smoking, bruxism, totally and extended partially edentulous with bone resorption that allowed insertion of standard size implants (length ≥ 10 mm), and absence of availability to participate in postoperative consultations.

This retrospective study was conducted by the principles of the Helsinki Declaration on Human Subject Experimentation. All patients included in this study signed consent, and the study was approved by the ethics council of “Stefan cel Mare” University, with the approval number 43/05.10.2021.

### 2.2. Preoperative and CBCT

Each of the patients in this study was initially subjected to orthopantomography (OPT). If the patient already had a complete mobile prosthesis, it was duplicated to create the diagnostic wax-up. Then, the dental technician prepared a resin prosthesis for scanning, which represents a copy of the diagnostic wax.

During the execution of cone-beam computed tomography, each of the patients needed to wear this scan prosthesis, with their mouth closed in occlusions. The CBCT investigation was also carried out for 3D evaluation of the bone. Finally, the scan prosthesis was scanned and saved as STL files (Figure 1).

The images were acquired from the Planmeca ProMax 3D Classic CBCT unit (Planmeca, Helsinki, Finland) using 90 kVp tube current, and 8 mA tube voltage. The voxel sizes of images were 200 μm, the field of view selection was Ø11 × 8.0 cm, and the focal spot was 0.5 mm. For visualization, we used the software Planmeca Romexis^®^.

### 2.3. Design and Fabrication of the Custom-Made Subperiosteal Implants

Custom-made implants were designed by 3D Medica SABETTIMED^®^ (Figure 2) and Bone Easy^®^ (Figure 3) with indications from the surgeon. In some cases, bone reduction was required to accommodate the bar, prosthetic components, and prosthesis. 

Each of the custom-made implants was designed with partial endosseous support to connect plates and suitable osseointegration. Implants were designed with a 0.7 mm thickness to adapt to both the jaw’s buttresses through fixation with 2 mm × 6 mm sandblasted large grits and acid-etched (SLA) treated osteosynthesis screws. Bone grafting was planned to be performed simultaneously with the placement of the implant.

The implant was manufactured by selective laser melting using a Truprint 1000 SLM machine, using Sintmill^®^ to place the implants on an indexation framework for posterior mechanization. 

After printing the base plate, the implants were fixed by supports and submitted to heat treatment—1 h heating to 800 °C stabilized for 30 min and cooling for 4 h. The frame and the implant were separated from the base and placed on a milling machine, using SUM 3D software to make M2 threads and re-mechanize the implant and the abutment connection. 

The plates were polished on the surface that directly contacts soft tissue, but the surface that contacts bone was left rough. The grafting zone was left unpolished. All devices were sterilized with Ethylene Oxide before surgery.

### 2.4. Surgical and Prosthetic Procedure

On the upper jaw, an incision was performed from each posterior region, with two relieving incisions in the posterior region. Buccal and palatal flaps were raised, exposing the anterior nasal spine, the pyriform apertures, the canine fossae, the zygomatic buttresses, and the posterolateral maxillae. The bone reduction was performed using a piezoelectric handpiece. Then, the implant was tested and fixed with osteosynthesis screws.

On the lower jaw, an incision was performed around the arch to the contralateral side. Anatomic landmarks such as external oblique ridges, mental foramina, mandibular symphysis, and genial tubercles were identified and exposed. A large bur was used to design the endosseous support. Then, the implant was tested and fixed with osteosynthesis screws (Figure 4, Figure 5, Figure 6 and Figure 7). 

Before ending the surgery, the abutments were placed, and flaps were closed with 4/0 vicryl^®^. Prosthetic impressions were taken immediately after closure, and within 12 h, the temporary fixed acrylic resin restoration was delivered to the patient. After 6 months following surgery, the final fixed restoration will be performed.

### 2.5. Outcome Measures

The measurements taken into account in this study were the fit and stability of implants, duration of each surgery, implant survival, and early and late complications.

Regarding the stability of each implant, a range from 0 to 5 was given by the surgeon during surgical intervention to evaluate the fit of DMLS subperiosteal implants to the corresponding bone anatomy. Scores of 0 to 2 indicated insufficient fit, 3 indicated a sufficient fit, and 4 to 5 were good to excellent fit.

The time it took for each intervention was monitored by a nurse, starting from the local anesthesia until the completion of the surgical intervention by making the suture. Time was measured in minutes and noted on the patient’s observation sheet. All subperiosteal implants that were functional one year after insertion were considered successful. 

Any immediate post-operative complications such as pain, swelling, edema, or bleeding occurring within 2 weeks after surgery were classified as early complications. 

Any complications that have occurred between the application of the first temporary prosthesis and the 6-month follow-up were classified as tardive complications. Tardive complications included suppuration, pain, swelling, implant exposure, implant mobility, and fracture of the temporary prosthesis. 

### 2.6. Statistical Analysis

All relevant patient data (gender, age at surgery) and implant and prosthesis information were gathered in an Excel spreadsheet. Means (±SD), ranges, medians, and confidence intervals (CI 95%) were calculated for quantitative variables (patients’ age), while absolute and relative (%) frequency distributions were obtained for all qualitative variables (patients’ gender). 

## 3. Results

Between August 2021 and January 2022, 16 totally or extended partially edentulous patients with marked bone resorption were considered for inclusion in this study for treatment with subperiosteal DMLS implants. Only one implant was lost, due to insufficient fit and recurrent, untreatable infections (Table 1). These aspects influenced the surgeon to remove the implant. Of all these 16 cases, 2 females were diagnosed with incipient osteoporosis, without bisphosphonates treatment. Eleven implants (69%) were placed on the maxillary and five (31%) implants were placed on the mandible (Table 1). 

Regarding the study results, the fit of the implants was extremely satisfactory with a mean rating of 4 out of 5. Only one implant (6%) had an insufficient fit, and this is the reason that this implant was lost (Table 1). Three of the implants (19%) had a sufficient fit, and twelve implants (75%) had a good to excellent fit (Table 1). On the maxillary, the fit of the implants had a mean rating of 3.81 out of 5, and on the mandible a mean rating of 4.4 out of 5.

The mean duration of the intervention was 86.18 min. However, this result was strongly influenced by the five cases in which the implant fit was not fully satisfactory, which required 113, 102, 105, 118, and 108 min from local anesthesia to suturing. The mean duration of surgical intervention on the maxillary bone was 90.27 min and 77.2 min on the mandible (Table 1). 

Regarding the early complications, pain, swelling, and edema were seen in the first three to five days after surgery for each patient, but just three patients experienced some of these complications 2 weeks after surgery. Another early complication was the bleeding, which was seen in three patients. 

The tardive complications, such as suppuration, pain, swelling, and implant mobility were absent in all of the patients. Another complication such as implant exposure was present in six patients. The grade of implant exposure was different in all six of these patients, but without affecting the functionality of the subperiosteal implant. One of the implants presented a fracture of the temporary prosthesis.

## 4. Discussion

Rehabilitation with different types of implants of severely resorbed jaws remains a challenge for oral and maxillofacial surgeons. There are many surgical protocols with good long-term reports for bone augmentation and endosseous dental implants. Although endosseous implants are associated with an increasing prevalence of peri-implantitis [16], subperiosteal implants represent an alternative solution for full jaw rehabilitation with immediate loading prosthesis.

The first subperiosteal implant was placed by G. Dahl in 1941 in the mandible of a patient in Sweden [19]. However, some of them showed complications such as implant exposure, mobility, and loss. In addition, the manufacturing technique for these types of implants was very rudimentary.

Gershkoff and Goldberg were the first authors which have been presented clinical cases with mandibular subperiosteal implants in the United States [15]. Leonard Linkowis performed many interventions with this type of implant which he followed for periods ranging from 2 to 12 years in several clinical studies [13,14,15].

The use of subperiosteal implants decreased in the late 1970s due to the appearance and increasingly frequent use of endosseous implants proposed by Branemark [14]. The reasons why subperiosteal implants disappeared were due to the fact that they required two surgical interventions: a first intervention in which the exposed bone was imprinted and the second with the application of the actual implant. Complicated procedures for making the actual implant and the rather high incidence of failures and complications resulted from it [15].

Currently, there is a digital revolution in medicine and dentistry related to new digital acquisition techniques, improved processing software, and modern manufacturing techniques, allowing the beginning of a new era for dental prosthesis, including the personalization of implant therapy [15,20,21,22].

One of the major complications of the first subperiosteal implants used was that these types of implants were non-rigidly fixed subperiosteal implants, and this led to progressive bone loss due to movement of the structure on the underlying bone, resulting in bone resorption. The new DMLS titanium subperiosteal implant uses a rigid fixation technique that is standard in trauma and reconstructive surgery today, which has been known since the 1950s [13].

Cerea and Dolcini [39] presented a series of seventy patients treated with custom direct metal laser sintering (DMLS) titanium subperiosteal implants that showed a 95.8% survival rate and low complication rates on a follow-up period of 2 years. In the conclusions of their study, they reported that customized subperiosteal DMLS implants could be a successful treatment technique for the prosthetic restoration of severely resorbed jaws when the placement of endosseous implants is not allowed [13]. The presented treatment solution for severe jaw resorption is innovative because it is customized to the patient’s anatomy and designed to include endosseous support. All implants were fabricated in rigid Ti6Al4V by SLM technology and fixed osteosynthesis screws treated with SLA. The authors believe that custom-made subperiosteal implants can be an excellent option for treating a severe atrophic jaw but the appropriate patients for this type of treatment would have to be chosen.

Our current study seems to positively associate with these results, as of 16 patients who were treated with custom DMLS titanium subperiosteal implants and then followed for a period of 6 months, a satisfactory implant survival rate was reported, with only one implant failure, due to insufficient fit and recurrent, untreatable infections. A low incidence of early postoperative complications was observed, and this aspect was probably a direct consequence of the reduced surgical time resulting from the excellent fit of custom implants. Early complications may consist of bacterial infections, possibly due to surgical technique, materials used, or even considering the ongoing COVID-19 pandemic with its health implications [40,41]. However, in our study, this was not the case. Over the 6-month follow-up period, the tardive complications were absent, except implant exposure which was present in six implants, but without affecting the functionality of this type of implant.

Furthermore, in our study, mostly all of the implants were indicated for a full arch; in just two cases, we treated a partial arch, and in one case, the implant did not survive.

Enough bone volume is required for the insertion of endosseous implants, and most of the time, the bone quantity is poor, especially in the posterior mandible of elderly patients. In addition, this region is one of the most difficult to regenerate with conventional procedures. Multiple bone regeneration procedures that have shown success in many cases involve more risks, with the appearance of complications that can increase the cost and duration of therapy.

Within the limits of the present study, the clinical application of custom-made DMLS titanium subperiosteal implants showed low complication rates and could become a solution for these patients. It is also very important to bring in discussions that the use of these types of implants should be performed only by specialists in Oral and Maxillofacial Surgery because of the extended surgery techniques that should be performed for these types of implants.

In this study, the main problems that could appear with this type of subperiosteal implants could relate to material fracture, infection, implant exposure, implant mobility, absence of osteointegration, and dimensions of the connection pillars used, which could predispose them to fractures of the implant and the prosthesis. Until now, in this study, we did not notice any of those possible complications with minor exceptions, such as small implant exposure in six patients and one temporary prosthesis fracture.

However, more long-term studies with larger samples of patients will need to be followed to further establish this technique.

## 5. Conclusions

Oral rehabilitation in patients with severe atrophy using a customized DMLS titanium subperiosteal implant could be a solution with great potential to combat the well-known problems of conventional implantology. It is also very important to have enough experience with implants in general, to reach the level of expertise for performing custom-made DMLS titanium subperiosteal implants.

## Figures and Tables

**Figure 1 diagnostics-12-02531-f001:**
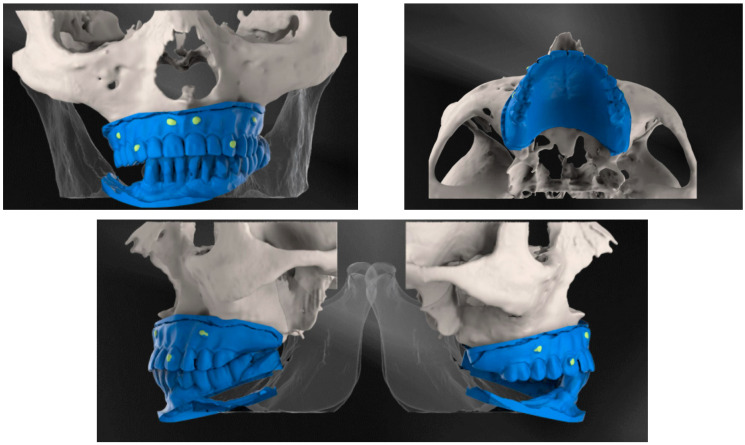
CBCT examination with 3D reconstructions with the maxillary prosthesis.

**Figure 2 diagnostics-12-02531-f002:**
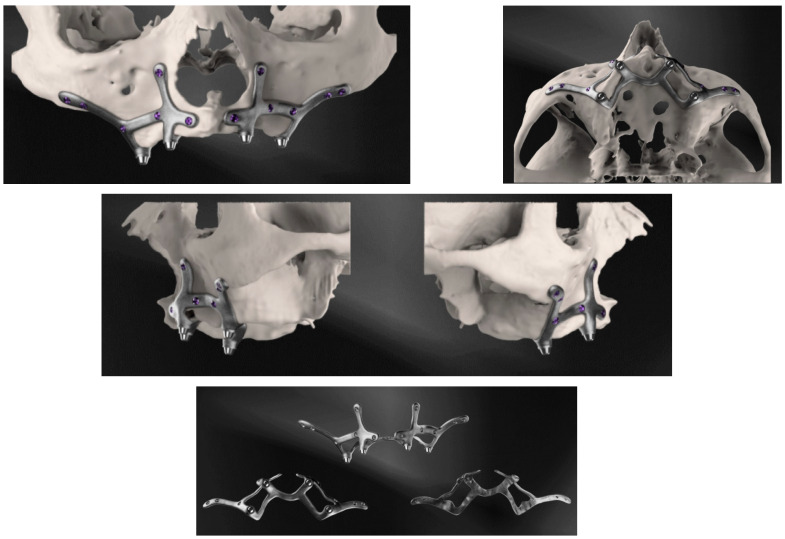
Three-dimensional reconstructions of the design of the subperiosteal implant for the upper jaw made by three-dimensional medical SABETTIMED^®^.

**Figure 3 diagnostics-12-02531-f003:**
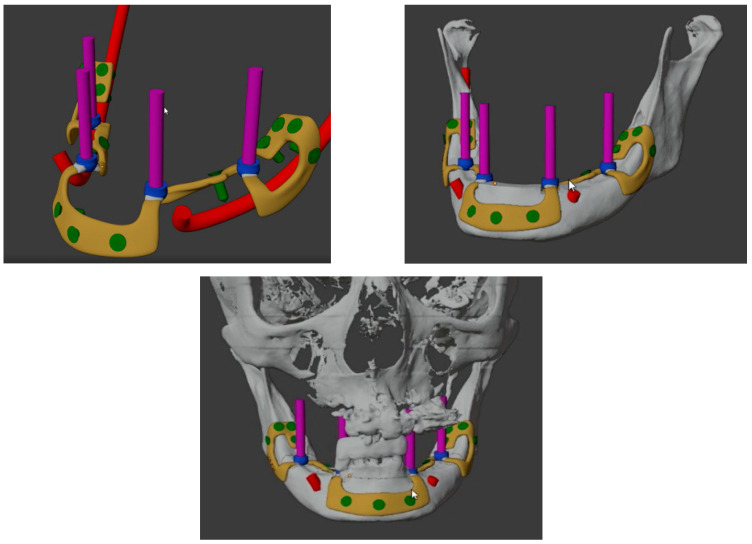
Three-dimensional reconstructions of the design of the subperiosteal implant for the lower jaw made by Bone Easy^®^.

**Figure 4 diagnostics-12-02531-f004:**
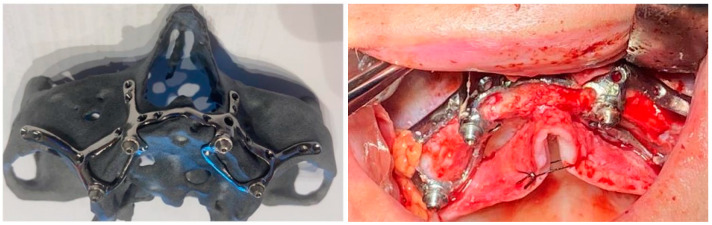
A patient with a total maxillary edentation treated with a subperiosteal implant (implant aspect and surgical stage).

**Figure 5 diagnostics-12-02531-f005:**
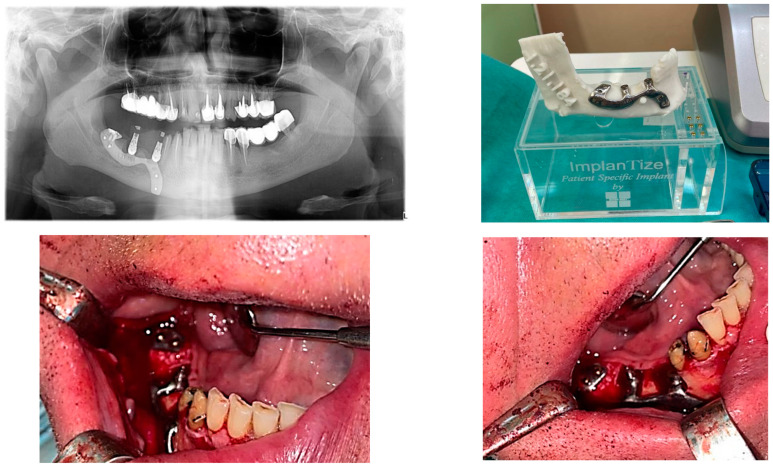
A patient with a terminal edentation on the right side of the mandible (missing teeth 4.5, 4.6, 4.7, 4.8) treated with a partially subperiosteal implant.

**Figure 6 diagnostics-12-02531-f006:**
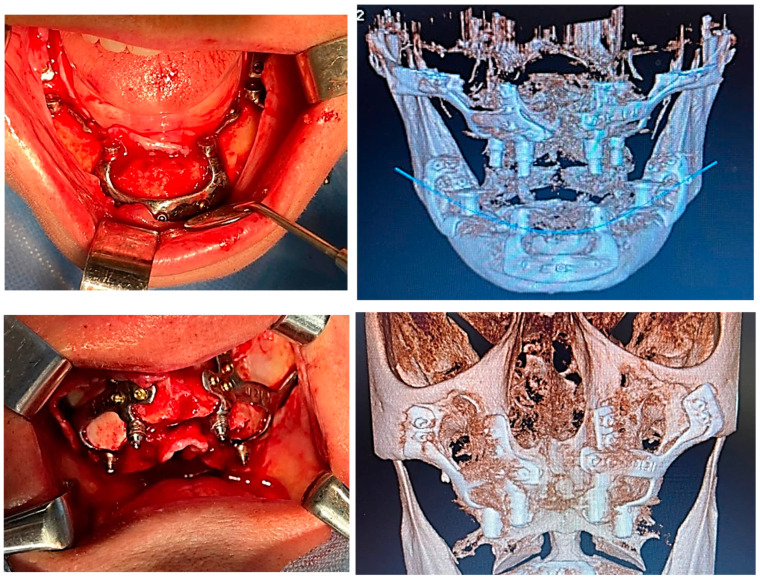
A patient with bimaxillary edentation treated with a subperiosteal implant in the maxillary and mandible (surgical stages and CBCT control after surgery).

**Figure 7 diagnostics-12-02531-f007:**
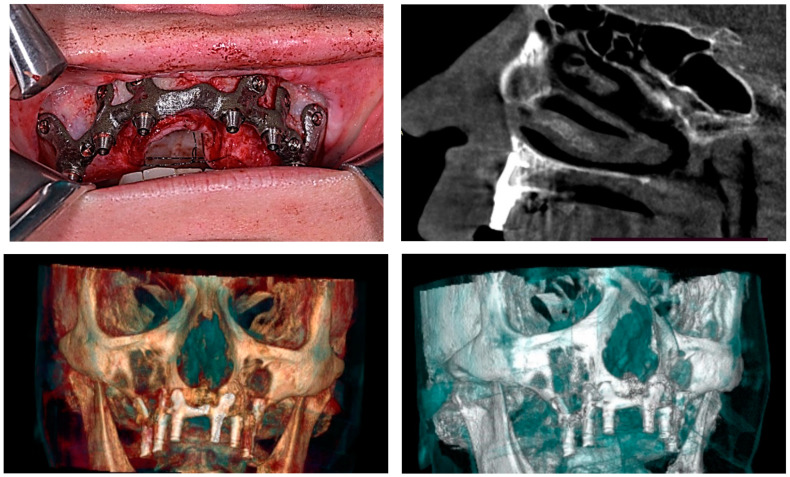
A patient with a total maxillary edentation treated with a subperiosteal implant (surgical stages and CBCT control after surgery).

**Table 1 diagnostics-12-02531-t001:** The study outcomes referred to patient and implant data.

Patients	Sex	Age	Localization/Type of Edentulism	Fit and Implant Stability	Duration of Intervention (Minutes)	Implant Survival
1	Male	68	Maxillary/full arch	5	53	yes
2	Female	57	Maxillary/full arch	5	51	yes
3	Male	68	Maxillary/partial arch	2	112	no
4	Female	67	Maxillary/full arch	4	67	yes
5	Male	67	Maxillary/full arch	3	102	yes
6	Male	56	Maxillary/full arch	4	105	yes
7	Male	59	Maxillary/full arch	3	118	yes
8	Female	55	Maxillary/full arch	4	97	yes
9	Male	61	Maxillary/full arch	5	89	yes
10	Male	62	Maxillary/full arch	4	91	yes
11	Female	55	Maxillary/full arch	3	108	yes
12	Male	63	Mandible/partial arch	4	95	yes
13	Female	56	Mandible/full arch	5	67	yes
14	Female	61	Mandible/full arch	4	78	yes
15	Male	56	Mandible/full arch	4	98	yes
16	Female	69	Mandible/full arch	5	48	yes

## Data Availability

Not applicable.

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
