# Peer review of "Custom-Made Direct Metal Laser Sintering Titanium Subperiosteal Implants in Oral and Maxillofacial Surgery for Severe Bone-Deficient Patients—A Pilot Study"

_diagnostics, 2022, doi:10.3390/diagnostics12102531_

Round 1
Reviewer 1 Report
Dear Authors,
The article: 'Custom-Made Direct Metal Laser Sintering Titanium Subperiosteal Implants in Oral and Maxillofacial Surgery for severe bone-deficient patients – a Pilot study' was to present a digital technique for the fabrication of custom-made subperiosteal implants and to report on the complication rates encountered when using these type of implants.
Start affiliations with the name of the department, then faculty, then university.
Punctuation mistakes should be corrected !!!!! English language and style are fine/minor spell check required.
Article must be prepared using MDPI guidelines (style etc).
The introduction is well written.
Materials and methods
Add information about CBCT (what equipment in detail)
Prepare the tables in accordance with the MDPI guidelines.
Figure 2 - Should be together on 1 page
Figures should be larger.
Results
Prepare the tables in accordance with the MDPI guidelines.
Figures should be larger.
Discussion is clearly presented.
Add table with abbeviations before references.
References should be prepared according MDPI guidelines.
Reconsider after major revision (control missing in some experiments)
Author Response
Dear Reviewer,
I had improved all the topics that you had recommended. I started with the affiliations and ended with the references which were written after the guidline.
Also, I introduced in the text about what CBCT tehnique we used for imaging and some new references.
Yours sincerely,
Alexandru Nemtoi
Reviewer 2 Report
The authors propose a method of implant treatment with Custom-Made Direct Metal Laser Sintering Titanium Subperiosteal Implants in Oral and Maxillofacial Surgery for severe bone-deficient patients.
The study is interesting and well organized however, there are some minor issues that must be addressed. Please see the enclosed PDF for further details.

Author Response
Dear Reviewer,
I had modified what you recommended in the text.
Yours sincerely,
Alexandru Nemtoi
Reviewer 3 Report
I have read with interest the paper “Custom-Made Direct Metal Laser Sintering Titanium Subperiosteal Implants in Oral and Maxillofacial Surgery for severe bone-deficient patients - a Pilot study”.
The subject considered is of interest, as little information is available in the literature on this newly introduced technique. Some improvements of the manuscript must be however suggested in order to provide adequate information:
- The patients considered are a little sample and from the images provided a number of variables may be identified, concerning the implant producer, the anatomical site (maxilla and mandible) the type of edentulism (full arch or partial); details on this data should be added in table 1, and discussed.
- Technical details on the CBCT scanning parameters should be provided, as this may affect artifacts and geometrical accuracy of virtually reconstructed bone structures.
- The use of “simultaneous bone grafting”, “bone reduction using a piezoelectric handpiece” and “large bur used to design the endosseous support” are mentioned to describe the surgical technique; as this complementary surgery may suggest limited accuracy of the implant fit, this aspect must be clarified and discussed.
- Different implant shapes and variables number of fixation screws and abutments are observed, thus explanation on the design method is mandatory.
- The surgical images may be redundant and could be reduced, while providing an example of clinica aspect at 1 year and of “implant exposition” could improve information.
- A relevant clinical aspect should be to provide data on probing, bleeding, inflammation around the abutments at 1 year follow-up.
- Text must be checked for spelling errors.
Author Response
Dear reviewer,
I modify the things that you recommended.
In the table 1 I introduced anothe parameter which is reffering to the type of edentulism: full or partial arch but i didn't introduce the name o f the producer because right now in this study I have just one implant without surviving and I do not want to influnece the readers if this article will be published. In future when we will have a greater sample of cases for sure I am going to publish more data, even the information about producer.
Thank you!
Yours sincerely,
Alexandru Nemtoi
Round 2
Reviewer 1 Report
Dear Authors,
Thank you for your corrections.
Article can be accepted after Editor decision.
Reviewer 3 Report
I congratulate the Authors for their study, and encourage submitting a larger sample with a longer follow-up.